# miR-665-Mediated Regulation of AHCYL2 and BVES Genes in Recurrent Implantation Failure

**DOI:** 10.3390/genes15020244

**Published:** 2024-02-15

**Authors:** Sung Hwan Cho, Young Myeong Kim, Hui Jeong An, Ji Hyang Kim, Nam Keun Kim

**Affiliations:** 1Department of Biomedical Science, College of Life Science, CHA University, Seongnam 13488, Republic of Korea; arana006@naver.com (S.H.C.); tody2209@naver.com (H.J.A.); 2College of Medicine, Konyang University, Daejeon 35365, Republic of Korea; 3Kangwon Institute of Inclusive Technology, Kangwon National University, Chuncheon 24341, Republic of Korea; ymkim@kangwon.ac.kr; 4College of Life Science, Gangneung-Wonju National University, Gangneung 25457, Republic of Korea; 5Department of Obstetrics and Gynecology, CHA Bundang Medical Center, School of Medicine, CHA University, Seongnam 13496, Republic of Korea

**Keywords:** recurrent pregnancy loss, transcriptome sequencing, small RNA sequencing, microRNA

## Abstract

The primary goal of this investigation was to identify mRNA targets affected by dysregulated miRNAs in RIF. This was accomplished by comprehensively analyzing mRNA and miRNA expression profiles in two groups: female subjects with normal reproductive function (control, n = 5) and female subjects experiencing recurrent implantation failure (RIF, n = 5). We conducted transcriptome sequencing and small RNA sequencing on endometrial tissue samples from these cohorts. Subsequently, we validated a selection of intriguing findings using real-time PCR with samples from the same cohort. In total, our analysis revealed that 929 mRNAs exhibited differential expression patterns between the control and RIF patient groups. Notably, our investigation confirmed the significant involvement of dysregulated genes in the context of RIF. Furthermore, we uncovered promising correlation patterns within these mRNA/miRNA pairs. Functional categorization of these miRNA/mRNA pairs highlighted that the differentially expressed genes were predominantly associated with processes such as angiogenesis and cell adhesion. We identified new target genes that are regulated by miR-665, including Blood Vessel Epicardial Substance (BVES) and Adenosylhomocysteinase like 2 (AHCYL2). Our findings suggest that abnormal regulation of genes involved in angiogenesis and cell adhesion, including BVES and AHCYL2, contributes to the endometrial dysfunction observed in women with recurrent implantation failure (RIF) compared to healthy women.

## 1. Introduction

Despite advances in assisted reproductive techniques (ARTs), many infertile couples continue to experience repeated treatment failure [1]. Recurrent implantation failure (RIF) is typically defined as the inability to achieve pregnancy after the transfer of a minimum of three high-quality blastocysts, as delineated by Shaulov et al. in 2020 [2]. However, there lacks a universally accepted definition, with variations in the definition of RIF influenced by factors such as maternal age, embryo quality, the inclusion or exclusion of aneuploidy screening, and the number of embryos transferred. The estimated incidence of RIF is approximately 10% among couples undergoing in vitro fertilization (IVF), according to Bellver and Simon [3].

Similar to recurrent pregnancy loss (RPL), a significant proportion, up to 50% of patients, experience unexplained RIF (uRIF) despite thorough investigations, as highlighted in research by Bashiri et al. [4]. The absence of a single, universally recognized definition underscores the complexity of RIF diagnosis and the challenge of addressing unexplained cases, emphasizing the need for further research and more comprehensive diagnostic criteria.

During embryo implantation, the endometrial window of implantation (WOI) undergoes a series of complex morphological and functional changes [5]. Recurrent implantation failure (RIF) correlates with alterations in the molecular and functional dynamics of endometrial receptivity during the WOI (days 20–24 of the cycle, during the secretory phase). Current investigations into the endometrial receptivity of RIF patients predominantly rely on transcriptomic signatures, leading to the identification of numerous molecules proposed as potential receptive biomarkers [6,7]. Nonetheless, the extensive array of biomarkers can exhibit variations among individuals, potentially leading to misleading interpretations of fertility status [8]. Hence, there is an urgent need to discern and establish specific key genes expressed in the endometrium to accurately assess its receptivity. Identifying these crucial genes could provide more reliable markers for evaluating endometrial receptivity in RIF patients, offering a more precise understanding of the factors influencing successful implantation. 

Recent advances in high-throughput sequencing have enabled comprehensive analysis of gene and protein expression patterns, providing new insights into the mechanisms of implantation failure [9,10,11]. Given the complexity of RIF, exploring its characteristics and pathogenesis from multiple perspectives using rapidly developing sequencing technologies is essential. This novel approach has the potential to result in more efficient therapeutic strategies for patients experiencing RIF.

Transcriptomics, such as microarray analysis or RNA sequencing (RNA-Seq), can provide a comprehensive overview of gene expression changes in the endometrium during the peri-implantation period. However, previous studies have primarily focused on the changing transcriptome profiles before and during the window of implantation [12,13].

To identify novel miRNAs and their targets associated with recurrent implantation failure (RIF), we performed transcriptome sequencing and small RNA sequencing on endometrial tissue. The schematic illustration of this study is shown in Figure 1. This study aimed to explore novel genes and miRNAs associated with recurrent implantation failure (RIF) by conducting transcriptome sequencing on endometrial tissues from both RIF patients and control groups. Following the sequencing analysis, statistically significant genes and miRNAs were identified. These findings facilitated the identification of miRNAs targeting the newly discovered genes. The comprehensive approach used in this study provides valuable insights into potential molecular mechanisms underlying RIF, offering a foundation for further investigations into targeted interventions for improving the outcomes of assisted reproductive techniques.

## 2. Materials and Methods

### 2.1. Ethics Statement

The study protocol was approved (IRB No. 2009-08-077) by the Institutional Review Board of CHA Bundang Medical Center, Seongnam, Republic of Korea. All study subjects provided written informed consent to participate in the study. The study abided by the Declaration of Helsinki and was approved by the Institutional Review Board of CHA Bundang Medical Center, CHA University. All procedures employed in the research adhered to the authorized protocols.

### 2.2. Subjects

The patients were enrolled from June 2017 through May 2018 at the Department of Obstetrics and Gynecology of CHA Bundang Medical Center (Seongnam, Republic of Korea).

The study’s RIF group consisted of patients who had undergone a minimum of three IVF cycles with high-quality embryos but had not achieved pregnancy (n = 5). RIF was defined as failure to achieve pregnancy after two or more completed fresh IVF-embryo transfer (IVF-ET) cycles with more than 10 cleaved embryos. Serum human chorionic gonadotrophin concentrations were less than 5 mIU/mL 14 days after embryo transfer. All transferred embryos were examined by the embryologist before transfer and judged to be of good quality. The criteria for identifying high-quality embryos in this study were determined by assessing specific parameters, such as morphological characteristics, developmental stage, and genetic normalcy. Hormonal causes of infertility, including hyperprolactinemia, luteal insufficiency, and thyroid disease, were excluded by measuring the concentrations of prolactin, thyroid stimulating hormone, free T4, follicle-stimulating hormone, luteinizing hormone, and progesterone in peripheral blood. Lupus anticoagulant and anticardiolipin antibodies were examined to identify the autoimmune diseases lupus and antiphospholipid syndrome, respectively. A thrombotic cause of RIF, namely thrombophilia, was evaluated by protein C and protein S deficiencies and by the presence of anti-a2 glycoprotein antibodies. Semen analysis, karyotyping, and oestradiol, testosterone, follicle-stimulating hormone, and luteinizing hormone assays were performed for male partners.

The control group (n = 5) was comprised of female volunteers below the age of 40 with regular menstrual cycles and a history of at least one uneventful pregnancy and childbirth. Participants with prior infertility issues and those currently using oral contraceptives or having intrauterine contraceptive devices were excluded. Transvaginal sonography was employed to examine the uterine cavities of both the control group and RIF patients and their endometrial thickness was measured. On the 21st day of their menstrual cycle (the mid-luteal phase), endometrial biopsies were collected using the Pipelle de Cornier1 device (CCD Laboratories, Paris, France). These biopsied samples were promptly transported to the research laboratory and snap-frozen at −80 °C for RNA extraction. 

### 2.3. RNA Sequencing

Transcriptome sequencing was performed using the Illumina Novaseq platform (Macrogen, Seoul, Republic of Korea). Sample libraries were independently prepared by Illumina TruSeq Stranded mRNA LT Sample Prep Kit (Illumina, Inc., San Diego, CA, USA, #RS-122-2101). The prepared libraries were subsequently sent for Illumina NovaSeq sequencing (Illumina, Inc., San Diego, CA, USA; conducted by Macrogen), and paired-end sequencing (2 × 100 bp) was carried out. The library protocol used the TruSeq Stranded mRNA Sample Preparation Guide, Part #15031047 Rev. E. Quality control was performed on the sequenced raw reads based on read quality, total bases, total reads, GC (%), and basic statistics. The quality of sequencing reads was assessed through a dual methodology, utilizing both FastQC for a comprehensive quality report and Phred quality scores for quantitative assessment of base call reliability within the sequencing data [14]. This combined approach ensured a comprehensive evaluation of data quality, providing confidence in the dataset’s integrity before initiating subsequent analyses. In our research, we acquired the Human genome reference (UCSC hg19) from NCBI and conducted read mapping to this reference. By utilizing HTSeq v0.10.0, we performed read counting and subsequently computed RPKM (reads per kilobase of transcript per million mapped reads) values to assess gene expression levels. This method facilitated a thorough analysis of gene expression using our sequencing data in conjunction with the reference genome. Following read mapping, we conducted transcript assembly using the StringTie program (StringTie version 1.3.4d, http://ccb.jhu.edu/software/stringtie/,version:stringtie-1.3.4d) (accessed on 16 May 2023). This yielded expression profiles for known transcripts in each sample, and we organized read counts and FPKM (Fragment per Kilobase of transcript per Million mapped reads) values based on transcripts and genes. Subsequently, for the requested comparative analysis (RIF_vs_Control), we utilized DESeq2 to perform differential expressed gene (DEG) analysis. To select differentially expressed genes between the two groups, 929 genes that satisfied the conditions of |fc| ≥ 2 and raw *p*-value < 0.05 were extracted.

### 2.4. Small RNA Sequencing

Endometrial RNA was extracted from endometrial tissue using the Trizol reagent (Invitrogen, Carlsbad, CA, USA) following the guidelines provided by the manufacturer. Small RNAs were sequenced using a TruSeq Small RNA Sequencing Kit (Illumina, San Diego, CA, USA) according to manufacturer instructions. Then, 1 μg total RNA was ligated with 3′ and 5′ adaptors, followed by reverse transcription with reverse transcriptase and amplification of the small RNA sequences during subsequent steps. This step converts RNA into complementary DNA (cDNA), preserving sequence information. The cDNA libraries are then amplified in a 15-cycle. The small RNA fraction, comprising the 130–150 base pair region, was excised from a 6% polyacrylamide gel post-electrophoresis, subjected to ethanol precipitation, and quantified. All samples were sequenced on an Illumina HiSeq 2500 System Sequencer using a 75-cycle High Output Kit. After sequencing, the raw sequence reads are filtered based on quality. Both trimmed reads and non-adapter reads as processed reads are used to analyze long targets (≥50 bp). Sequenced reads are classified as trimmed reads, non-adapter reads, short reads, and low-quality reads according to these criteria. The adapter trimming process is performed to eliminate the adapter sequences that exist in the read using Cutadapt (https://cutadapt.readthedocs.org/en/stable/, accessed on 16 May 2023). 

FastQC was used to perform quality checks on the raw sequences prior to analysis to ensure data integrity (http://www.babraham.ac.uk/projects/fastqc/, accessed on 16 May 2023). miRbase was used to search and explore predicted hairpin sequences and maturation sequences of miRNA transcripts (http://www.mirbase.org/, accessed on 16 May 2023). The DEGs with the threshold of |fc| ≥ 2 and raw. *p* < 0.05 were analyzed via DESeq2 (http://www.bioconductor.org/packages/release/bioc/html/DESeq2.html, accessed on 16 May 2023). The target genes of miRNAs were predicted by TargetScan (http://www.targetscan.org/mamm_31/, accessed on 16 May 2023), miRBase (http://www.mirbase.org/, accessed on 16 May 2023), miRTARBase (http://mirtarbase.cuhk.edu.cn/php/index.php, accessed on 16 May 2023), and miRNet (https://www.mirnet.ca/upload/MirUploadView.xhtml, accessed on 16 May 2023). 

### 2.5. Gene Ontology Enrichment Analysis

We performed the Kyoto Encyclopedia of Genes and Genomes (KEGG) pathway enrichment analysis and Gene Ontology enrichment analysis (biological processes) to identify the functions of DEGs by using the EnrichR database (http://amp.pharm.mssm.edu/Enrichr/, accessed on 16 May 2023). The Database for Annotation, Visualization, and Integrated Discovery (DAVID) (https://david.ncifcrf.gov/, accessed on 16 May 2023) is a database for differential analysis of genes, enrichment of pathways, and association to biological annotation. Top enriched terms were generated according to the lowest *p*-value < 0.05 (Fisher’s exact test). The molecular function and protein class related to the angiogenesis and cell adhesion were performed in the PANTHER classification system v. 11.0 (http://www.pantherdb.org/, accessed on 16 May 2023). Heat maps and scatter plots for clustering analyses were performed using the web tool Morpheus (https://software. broadinstitute.org/morpheus, accessed on 16 May 2023).

### 2.6. Validation of Candidate Gene Expression

To validate the candidate gene expression from RNA sequencing, we used real-time qRT-PCR. Total RNA (500 ng) was used for RT reactions that were performed using the miScript II RT kit (Qiagen, Hilden, Germany) according to the manufacturer’s protocol. Thaw all RNA on ice. Thaw 10× miScript Nucleics Mix and 5× miScript Hiflex Buffer at room temperature (15–25 °C). Mix each solution by mixing the tubes well. Then, centrifuge briefly to collect residual liquid from the sides of the tubes and store it on ice. Add RNA to each tube containing reverse-transcription master mix (total 20 microliters). Incubate at 37 °C for 60 min for amplification. Real-time PCR was performed on the ABI StepOnePlus™ system (Foster City, CA, USA). For the quantitative analysis of predicted target genes, the primers were designed using Primer Premier 3 software. *GAPDH* RNA was used as an endogenous control. All primers were part of SYBR green assays for target genes or *GAPDH* (Bioneer, Daejeon, Republic of Korea). For each target sequence, we determined the cycle number at which the product level exceeded an arbitrary Ct (Cycle threshold). Additionally, the relative amount of each target gene in relation to *GAPDH* RNA was calculated using the formula 2-deltadeltaCt. The PCR conditions were as follows: 95 °C for 5 min, followed by 40 cycles of 95 °C for 15 s, 58 °C for 15 s, and 72 °C for 30 s. Reactions were run in duplicate using RNA samples from three independent experiments. The fold change in expression of each gene was calculated using the 2-deltaCt (deltaCt, relative cycle threshold compared with *GAPDH*) method, with *GAPDH* as an internal control. 

### 2.7. Validation of Candidate miRNAs and Expression of Their Target Gene

To evaluate miR-665, miR-4732-5p, and miR-375 expression, real-time RT-PCR was used. RNA (500 ng) was used for RT-PCR reactions that were performed using an miScript II RT kit (Qiagen, Hilden, Germany) according to the manufacturer’s protocol. Thaw all RNA on ice. Thaw 10× miScript Nucleics Mix and 5× miScript HiSpec Buffer at room temperature (15–25 °C). Mix each solution by mixing the tubes well. Then, centrifuge briefly to collect residual liquid from the sides of the tubes and store it on ice. Add RNA to each tube containing reverse-transcription master mix (total 20 microliters). Incubate at 37 °C for 60 min for amplification. When reverse transcription reactions are performed, mature miRNAs, as well as certain small nucleolar RNAs and small nuclear RNAs, are selectively converted into cDNA. Mature miRNAs are polyadenylated by poly(A) polymerase and reverse transcribed into cDNA using oligo-dT primers. The oligo-dT primers have a 3′ degenerate anchor and a universal tag sequence on the 5′ end, allowing amplification of mature miRNA at the real-time PCR step. Real-time PCR was performed on the Roter-Gene System. *RNU6* RNA was used as an endogenous control. All primers were part of SYBR green assays for miR-665, miR-4732-5p, and miR-375 or RNU6 (Qiagen, Hilden, Germany). For each target sequence, the number of cycles in which the cDNA amplification level arbitrarily exceeded Ct (cycle threshold) was determined and compared using RNU6 RNA as a normalization control, and the amount of each miRNA was quantified using the 2-delta-deltaCt formula. For the quantitative analysis of target genes, the primers were designed using Primer Premier 3 software. *GAPDH* RNA was used as an endogenous control. Amplification of the target gene or *GAPDH* was performed using the SYBR green detection system, and the SYBR green mixture (Bioneer, Daejeon, Republic of Korea) was used for the RT-PCR reaction. The cycle number at which the product level exceeded an arbitrary Ct (cycle threshold) was determined for each target sequence, and the amount of each target gene relative to *GAPDH* RNA was described using the formula2-deltadeltaCt. To identify the target genes of miRNAs, experimentally validated miRNA and target gene pairs were retrieved from MirTarBase and miRNet, respectively. The mature miRNA sequences were downloaded from the miRBase database. 

### 2.8. Expression Vector Construction

To synthesize 3′-UTR of *AHCYL2* and 3′UTR of *BVES*, fragments (3′-UTR of *AHCYL2*, 1326 bp and 3′UTR of *BVES*, 1057 bp) were amplified from human genomic DNA with primers (3′-UTR of AHCYL2, forward: 5′-CCTTTTGCCTACCTAGTCCTGA-3′, reverse: 5′-TGTTGCCTATTTTCCTTCCGT-3′; 3′UTR of BVES, forward 5′-TCAGAGAGAGAATTCAGGTTAC-3′, reverse: 5′-TCAGAGCAGTTTATAAATGGCA) and cloned into the vector pmiRGLO (Invitrogen, Carlsbad, CA, USA)-3′-UTR of AHCYL2 with a XhoI (CCGCTCGAG)/XbaI (CTAGTCTAGA) linker, and the vector pmiRGLO (Invitrogen, Carlsbad, CA, USA)- 3′-UTR of BVES with a NotI (GCGGCCGC)/XbaI (CTAGTCTAGA) linker. The sequences of constructs were confirmed by direct sequencing. miR-665 (miR0513-MR04) and Has-miR-665 inhibitor (HmiR-AN0778-AM02), a target miRNA corresponding to the 3’-UTR region of AHCYL2 and BVES, were purchased from Genecopoeia (Rockville, MD, USA).

### 2.9. Cell Transfection and Luciferase Assay

The endometrial adenocarcinoma Ishikawa cells and Hela cells were plated at 3 × 10^5^ cells per well in a 6-well plate and transfected 24 h later using Lipofectamine 3000 transfection reagent (Thermofisher, Waltham, MA, USA). Each transfection reaction contained 500 ng of miR-665 (in pEZX-MR04) and miR-665 inhibitor (in pEZX-AM02) for 3′UTR of *AHCYL2* and 3′UTR of *BVES* along with 3′-UTR-AHCYL2 in pmiRGLO, 3′-UTR-BVES in pmiRGLO, and pmiRGLO (control). The Dual-Luciferase Reporter Assay System (DLR assay system, Promega, Madison, WI, USA) was used to perform dual-reporter assays on pmiRGLO-based reporter systems. The DLR assay system was employed to quantify the luciferase activity of cells co-transfected with the miR-665 expression vector and the 3′-UTR of the target gene inserted into pmiRGLO. Twenty-four hours after transfection, the growth medium was removed, and cells were washed once or twice with phosphate-buffered saline. Passive lysis buffer (200 µL/well; Promega, Madison, WI, USA) at 200 µL/well was added, and the plates were rocked gently for 15 min at room temperature, after which cell lysates were harvested for DLR assay. Cell lysates (20 µL) were transferred into white opaque 96-well plates (Corning Inc., Corning, NY, USA). Then, we conducted sequential firefly and Renilla luciferase activity assays on the cell lysate in each well. Relative luciferase activity was calculated as the ratio of Firefly/Renilla luciferase activity (transfected with pmiRGLO vector as internal control) to normalize cell numbers and transfection efficiency.

### 2.10. Statistical Analysis

Statistical analysis was performed using Prism software (GraphPad software, Inc, La Jolla, CA, USA). All statistical analyses were performed using an unpaired two-tailed Student’s *t*-test as applicable. When applicable, data were displayed as mean ± SEM. *p*-values < 0.05 were considered statistically significant.

## 3. Results

### 3.1. Gene Expression Profiles of the Endometrial Tissue with RIF Are Distinctly Different from Those of Healthy Control

To identify novel miRNAs and their targets associated with recurrent implantation failure (RIF), we performed transcriptome sequencing and small RNA sequencing on endometrial tissue. The schematic illustration of this study is shown in Figure 1. To establish differential gene profiling for RIF, we first analyzed mRNA information for RIF patient samples by transcriptome sequencing. Compared with the control sample, RIF patients’ samples exhibited a total of 929 differentially expressed genes, including 340 upregulated genes and 589 downregulated genes (Figure 2A). This list included top10 genes (*GDF10*, *S100A9*, *ASTN1*, *COL26A1*, *CYP24A1*, *SCARNA13*, *PDZK1P1*, *PDZK1*, *FAM129C*, and *USP32P1*) (Appendix A) that showed most upregulated genes between the RIF and control groups and adjusted *p*-values less than 0.05 in both the populations and Top10 genes (*SLC24A4*, *PHF24*, *XDH*, *HLA-DOB*, *MEGF10*, *TRPM6*, *KCNG1*, *PKHD1L1*, *LOC101928150,* and *CAPN6*) (Appendix A) that showed at least a two-fold downregulation between the RIF and control groups and adjusted *p*-values less than 0.05 in both the populations. Scatter plots and volcano plots were used to display the candidate genes based on their expression levels compared between samples (Figure 2B,C).

### 3.2. Gene Set Enrichment Analysis

GO enrichment analysis showed that DEGs were enriched in biological processes (BP), cellular components (CC), and molecular functions (MF). Figure 2D–F presents the GO terms of the DEGs ranked according to their enrichment factor, which presents data regarding the top 10 GO terms. Of the GO terms in the category biological process, ‘single-organism process’ and ‘cellular process’ had the highest enrichment factors, ‘cell’ and ‘cell part’ had the highest enrichment factor in the category CC and ‘binding’, and ‘protein binding’ had the highest enrichment factor in the category MF. 

### 3.3. Validation of Differential Expressed Genes

Real-time PCR (RT-qPCR) was performed using endometrial tissue (ET) to verify the gene RNA sequencing results for selected DEGs. Of the endometrial tissue–differential expressed genes (ET-DEGs), the expression levels of 10 randomly selected genes were statistically significantly different [adjusted *p* ≤ 0.05] between RIF and the corresponding controls (Figure 3). As presented in Figure 3, the expression levels of *RRM2*, *ST3GAL5*, *TGFB2*, *SAMD11*, *SMPDL3B*, and *FAX* (Figure 3A–F) were higher in ET obtained from RIF patients compared with those in ET obtained from healthy controls while the expression levels of *ADK*, *MICB*, *CSRP2*, and *MYO10* (Figure 3G–J) were lower in ET obtained from RIF patients compared with those in ET obtained from healthy controls (Figure 3). Primer information for the candidate and reference genes are listed in Table 1. 

### 3.4. Identification of Differentially Expressed miRNAs

We performed miRNA sequencing in the endometrial tissue samples derived from patients with RIF (n = 5) and successful clinical pregnancy (n = 5) to identify the differential miRNAs. To uncover previously unidentified miRNAs within our sequencing dataset, we aligned the clean miRNA reads with the human genome and employed miRDeep2 (version 2.0.08), an algorithm grounded in the miRNA biogenesis model. A total of 1378 miRNAs were analyzed through small RNA sequencing, and among them, 98 upregulated miRNAs and 88 downregulated miRNAs were selected based on significance tests (significant RandFold *p*-values, *p* < 0.05). Twenty-four miRNAs (14 upregulated, 10 downregulated) met the criteria of fold change (|fc| ≥ 2) and raw *p* < 0.05 using DESeq2. We used the online tools miRTarBase and miRnet to pinpoint experimentally validated targets supported by strong evidence. From these findings, we identified seventeen miRNAs demonstrating a negative correlation with the candidate genes (Appendix A). 

As a result, we identified a set of differentially expressed miRNAs (DEmiRs) comprising ten upregulated (hsa-miR-3667-3p, hsa-miR-6858-3p, hsa-miR-216a-3p, hsa-miR-410-5p, hsa-miR-106a-3p, hsa-miR-6814-3p, hsa-miR-1224-3p, hsa-miR-4732-5p, hsa-miR-129-2-3p, and hsa-miR-665) and seven downregulated (hsa-miR-30d-5p, hsa-miR-3188, hsa-miR-375, hsa-miR-6894-5p, hsa-miR-1913, hsa-miR-4743-5p, and hsa-miR-3663-3p), demonstrating high reliable miRNAs in RIF (Appendix A). 

A small RNA sequencing heat map was constructed to depict the expression profiles of the top 24 genes exhibiting the most significant increases or decreases in response to RIF. The color spectrum from blue to yellow indicates low to high expression (Appendix A). The MA plot shows the average fold change vs. the average expression level of these DEmiRs (Appendix A). The scatter plot shows the upregulated and downregulated genes (the orange and red dots, respectively) in the endometrial tissue in the RIF(PR) concerning the control (CR). Scatter plot gray dots represent genes with no significant difference (Appendix A). The volcano plot (*p*-value vs. linear fold change) of all genes present in mRNA sequencing shows the comparison between RIF patients and controls. Fold change (FC ≥ 2) with yellow indicates upregulation in RIF; linear fold change (l FC ≥ 2) with blue indicates downregulation in RIF. The X-axis shows the linear FC level, and the Y-axis shows the significance level (Appendix A). miRNA sequencing provided a list of 186 differentially expressed miRNAs that overlapped between the criteria (|fold change| ≥ 2) (Appendix A). miRNA sequencing provided a list of 24 differentially expressed miRNAs that overlapped between the criteria (|Fold-Change| ≥ 2 fold, *p* ≤ 0.05) (Appendix A).

### 3.5. Inverse Correlation of miRNAs and Putative Target Genes in the ET of Control and RIF

The availability of miRNA-Seq and mRNA-Seq data from both RIF and control samples is a substantial advantage. This enables the investigation of the negative correlation between miRNA and mRNA expressions, a prerequisite for establishing meaningful miRNA–mRNA target relationships. To identify the mRNAs targeted by miRNAs, we predicted miRNAs target mRNAs based on miRTarbase, miRnet, and targetscan. As a result, we identified 17 specific miRNAs targeting 71 candidate genes (Appendix A).

As a result, we identified the expression of miR-665, miR-4732-5p, and miR-375 in endometrial tissue that correlated with the candidate genes (Figure 4). To validate miRNAs targeting candidate genes identified through transcriptome sequencing and small RNA sequencing, we performed qRT-PCR on endometrial tissue obtained from patients experiencing RIF. The expressions of miR-665 and miR-4732-5p (Figure 4A,G) were significantly higher in RIF compared to the control. Additionally, the expressions of *NRXN3*, *BVES*, *PLA2G4A*, *AHCYL2*, and *MICA* (Figure 4B–F), putative target genes of miR-665, and *PRKX* and *CNKSR3* (Figure 4H,I), putative target genes of miR-4732-5p, were significantly lower in RIF compared to the control. Conversely, the expression of miR-375 (Figure 4J) was significantly lower in RIF compared to the control. Moreover, the expressions of *TGFB2* and *ARHGAP11A* (Figure 4K,L), putative target genes of miR-375, were relatively higher in RIF. The relative levels of miR-665, miR-4732-5p, and miR-375 were normalized to *U6*, and the relative levels of target genes were normalized to *GAPDH*. ** *p* < 0.05. The primer information is listed in Table 2.

### 3.6. AHCYL2 and BVES Are Direct Targets of miR-665 in Ishikawa Cells

To identify potential factors regulating *AHCYL2* and *BVES* expression in RIF disease, we performed transcriptome sequencing, small RNA sequencing, and bioinformatics analysis to identify miR-665 as a potential target of the *AHCYL2* and *BVES* genes. We found that *AHCYL2* and *BVES* 3′-UTR contain the miR-665-validated binding target site and then used the vectors (pmirGLO vector) of human *AHCYL2* and BVES fused downstream of the firefly luciferase gene. Reporter gene assay demonstrated that *AHCYL2* and *BVES* mRNA levels were significantly suppressed after miR-665 overexpression in Ishikawa cells and Hela cells (Figure 5). A schematic diagram of a gene with a 3′-UTR of *AHCYL2* and *BVES* containing possible miR-665 binding sites in a conserved region is shown in Figure 5a,b. The results from the luciferase reporter assay showed that the luciferase activity was significantly decreased in Ishikawa cells and Hela cells (Figure 5c,d) and Hela (Figure 5e,f) cells after co-transfection of miR-665 expression vector with 3′-UTR of *AHCYL2* and *BVES* reporters, which was negatively effected by the off-target miRNA in the putative miR-665 binding site. The sequence information is listed in Appendix A.

## 4. Discussion

The multifactorial pathogenesis of RIF remains incompletely understood, but known risk factors include age, BMI, tobacco and alcohol consumption, and a history of endometriosis. Male factors (sperm quality) and female factors (thrombophilia, vitamin D deficiency, etc.) have also been implicated in RIF development [15]. Transcriptomics, specifically RNA sequencing, offers a comprehensive overview of the genetic expression alterations within the endometrium during the peri-conception period, presenting valuable insights into the dynamics of gene expression changes. Our investigation initially uncovered 24 differentially expressed miRNAs (DEMs) and 929 differentially expressed genes (DEGs) by analyzing miRNA and mRNA expression profiling datasets, respectively, comparing RIF sample tissues to healthy endometrial tissues. Among the DEMs, 14 were upregulated, and 10 were downregulated miRNAs, whereas the DEGs comprised 340 upregulated genes and 589 downregulated genes. From the pool of identified genes and miRNAs, we selected a total of 71 genes and 17 miRNAs, putatively targeting these genes for further analysis. In particular, miR-375, miR-665, and hsa-miR-4732-5p showed interesting inverse correlations with target genes in this selected cohort, making them prime candidates for further study. Ten of these genes (*NRXN3*, *TGFBR2*, *KSR2*, *AHCYL2*, *PLA2G4A*, *MICA*, *BVES*, *TGFB*, and *ARHGAP11A*) are known to be involved in processes related to angiogenesis (the formation of new blood vessels) and cell adhesion (cells sticking together). Because these genes play an important role in RIF, we included them in the cohort selected for further analysis. 

Angiogenesis, the formation of new blood vessels, plays a pivotal role in establishing the necessary vascular network within the endometrium to support successful embryo implantation [16]. The intricate balance of pro-angiogenic and anti-angiogenic factors within this microenvironment is crucial for creating an optimal milieu for implantation. Numerous studies have linked alterations in angiogenesis and vascular-related genes with recurrent implantation failure (RIF), emphasizing their potential significance in the pathophysiology of this condition. Research conducted by Von Grothusen (2022) [17] shed light on the dysregulation of vascular-related genes in th endometrial tissues of RIF patients. This study identified downregulation of pro-angiogenic factors, such as *VEGF* (Vascular Endothelial Growth Factor), and an upregulation of anti-angiogenic factors, like *TSP1* (Thrombospondin-1), indicating an imbalance in the angiogenic milieu. Their findings indicated that certain genetic variations in angiogenesis-related genes, particularly in the *VEGF* pathway, were significantly associated with an increased risk of RIF among specific patient cohorts. Women with infertility or recurrent miscarriages have lower levels of endometrial *VEGF* expression [18,19] and peripheral blood levels of *MMP-7* and *VEGF*, suggesting that angiogenesis plays a role in implantation and placenta development [20]. These molecules could be potential biomarkers for and therapeutic targets of RIF [21,22]. Additionally, extensive angiogenesis and vasculogenesis occur in both maternal and fetal placental tissues, and the fibroblast growth factor (FGF) family of molecules and their receptors are potent mediators of angiogenesis in the placenta [23]. Understanding the dynamics of angiogenesis and the role of vascular-related genes in the context of RIF is fundamental for developing targeted interventions that could potentially improve endometrial receptivity and increase the success rates of assisted reproductive techniques. 

Cell adhesion plays a crucial role in the process of embryo implantation. Embryo implantation marks the first direct cellular interaction between the mother and the developing embryo. This process consists of a complex and coordinated series of events, and human embryo implantation requires three sequential stages: apposition, adhesion, and invasion [24]. Cell adhesion is the second stage of this implantation process, where the newly hatched embryo needs to be slightly attached to the epithelium of the uterine lining. This stage often occurs as the embryo “rolls” to its final location for implantation. Subsequently, the embryo firmly attaches to the uterine lining [25]. Therefore, cell adhesion is an essential process for the embryo to successfully attach to the uterine lining and for pregnancy to occur. If this process does not occur properly, embryo implantation may fail. For these reasons, cell adhesion is closely related to embryo implantation.

Among the genes obtained from the endometrial tissue of RIF patients that we analyzed, numerous genes related to angiogenesis have been identified. Of these genes, *NRXN3*, a human gene, influences angiogenesis, a process involving blood vessel formation [26]. NRXN3’s role in angiogenesis may be linked to its influence on gene expression related to blood vessel growth [27]. The protein may impact an angiogenic state characterized by increased expression of genes involved in angiogenesis and antigen presentation [26]. *AHCYL2* is a gene associated with various biological processes, including angiogenesis, as indicated in several studies [28,29]. This gene has been linked to tumors, specifically showing associations between *AHCYL2* and certain cancers. Studies have revealed correlations between *AHCYL2* expression and tumor cell proliferation, angiogenesis, and poor prognosis in some cancers [30]. Additionally, DNA methylation patterns of *AHCYL2* have been proposed as potential diagnostic and prognostic markers [31]. *PLA2G4A* is a gene encoding cytosolic phospholipase A2, which catalyzes membrane phospholipid hydrolysis [32]. Research links it to tumor progression and angiogenesis, notably in brain and lung cancers [33]. Studies suggest its overexpression is associated with tumorigenesis and angiogenesis, particularly in basal-like breast cancer [34]. Antibodies targeting VEGFR2 fused with MICA stimulate antiangiogenic effects, showing clinical benefits in cancer therapy, particularly in gastric and lung cancers [35]. Li et al. (2011) provided the initial evidence demonstrating the stimulating role of PRKX in angiogenesis [36]. Their study illustrated the critical involvement of PRKX in regulating all three crucial processes necessary for angiogenesis: proliferation, migration, and the formation of vascular-like structures in endothelial cells. This protein is involved in crucial physiological events like embryonic development and wound repair. *CNKSR3*, a gene, shows ubiquitous nuclear expression associated with various cell functions. It is linked to angiogenesis, the formation of new blood vessels from pre-existing ones [37]. BVES plays a crucial role in regulating adhesion between cells. It is involved in maintaining cell–cell interactions, influencing cell shape, and directing normal cellular processes [38]. Adhesion proteins like BVES are critical during processes such as blastocyst implantation in the uterus, as they facilitate the attachment of the embryo to the maternal endometrial lining [39]. *TGFB2*, a gene associated with the transforming growth factor-β (TGF-β) signaling pathway, plays a crucial role in angiogenesis, the formation of new blood vessels. Impaired TGFB2 function can lead to reduced branching and inhibition of endothelial cell migration, affecting proper angiogenesis [40]. In various pathological conditions, ARHGAP11A emerges as a multifaceted regulator involved in cancer cell mobility, Aβ neurotoxicity, and angiogenesis, making it a potential therapeutic target [41]. 

Meanwhile, as regulators of target genes, microRNAs (miRNAs) have been implicated in complex processes related to transplantation, particularly recurrent graft failure (RIF). Studies have suggested that miRNAs play a crucial regulatory role in various stages of implantation, affecting endometrial receptivity, embryo development, and the dialogue between the embryo and the endometrium. A study by Goharitaban et al. (2018) [42] discussed the dysregulation of specific miRNAs in RIF patients, highlighting their potential impact on key pathways involved in implantation failure. Additionally, the work by Suh et al. (2015) [43] identified several miRNAs linked to the regulation of genes critical for successful implantation, shedding light on their significance in this process. This evidence underscores the importance of miRNAs as potential regulators in the context of implantation, especially in the pathological condition of RIF. Further exploration of the specific miRNA profiles and their target genes could offer insights into understanding and potentially addressing implantation failure. Several microRNAs have been identified in the context of recurrent implantation failure (RIF), including miR-145, miR-let-7 family, miR-30 family, miR-223, and miR-126. These miRNAs are implicated in influencing various aspects of the implantation process, such as endometrial receptivity, immune response, and vascular development, underscoring their potential as regulators in RIF [44,45]. Further investigation into their specific mechanisms and clinical applications is warranted.

In the context of implantation, microRNAs (miRNAs) emerge as pivotal regulators influencing critical processes such as decidualization and angiogenesis. Regarding angiogenesis, miRNAs like miR-126 and miR-210 exert influence over angiogenesis, contributing to the formation of new blood vessels—a vital aspect of implantation. These miRNAs regulate factors involved in vascular development and endothelial cell function [46]. Understanding the regulatory roles of miRNAs in these processes is crucial as they significantly impact the success of implantation. Their involvement in governing cell proliferation, orchestrating the transformation of the endometrium for embryo acceptance, and facilitating the establishment of an adequate vascular network underscores their importance in the intricate series of events necessary for successful implantation.

To our knowledge, the association of three newly identified miRNAs (miR-375, miR-665, and hsa-miR-4732-5p) with recurrent implantation failure (RIF) has not been reported to date. Hsa-miR-665 demonstrates significant involvement in the modulation of cell proliferation and apoptosis by influencing the Wnt5a/β-catenin signaling pathway, promoting cell proliferation, and regulating cell cycle transitions [47]. hsa-miR-4732-5p is an identified microRNA (miRNA) found in *Homo sapiens* involved in various biological processes. It is particularly significant in diseases like ovarian cancer [48]. This miRNA promotes ovarian cancer mobility and cell viability. Weight was found to be positively associated with miR-375. Previous studies have demonstrated that miR-375 regulates insulin exocytosis and is associated with type 2 diabetes [49]. Infertility is also linked to abnormal lipid levels, obesity, and metabolic disorders [50], and miRNAs have been reported to connect infertility with metabolism [51].

Although the newly identified genes from transcriptome data have yet to be reported in association with RIF, genes related to angiogenesis and cell adhesion, such as *NRXN3*, *BVES*, *TGFB2*, *AHCYL2*, *PLA2G4A*, *PRKX*, *CNKSR3*, and *MICA*, play a crucial role in facilitating endometrial vascular development to ensure adequate blood supply for supporting embryo growth. When these genes are suppressed by miRNAs, their diminished expression may disrupt the intricate process of vascular neogenesis, potentially impacting the microenvironment crucial for facilitating successful embryo implantation.

In our study, we have observed significant changes in the expression of genes related to angiogenesis and cell adhesion in the context of recurrent implantation failure (RIF). Particularly noteworthy are the findings related to the *BVES* and *AHCYL2* genes, where our data indicate that their expressions are regulated by miR-665. The dysregulation of these genes, resulting from miRNA-mediated downregulation, is believed to significantly impact the coordination and integration of the endometrial vascular system. The identified miRNA-mediated downregulation of *AHCYL2* and *BVES* genes is of particular interest due to their critical roles in angiogenesis and cell adhesion. These processes are integral for the establishment of a conducive environment for embryo implantation. The disruption in the normal expression patterns of these genes, as we have observed in RIF, can compromise the endometrial receptivity essential for successful implantation. The downregulation of *AHCYL2* and *BVES* genes, orchestrated by miR-665, implies a regulatory mechanism that directly influences the endometrial vascular system. This dysregulation can negatively impact vascular development, potentially leading to impaired support for the developing embryo. Understanding the specific miRNA-mediated regulatory pathways affecting angiogenesis and cell adhesion genes provides valuable insights into the molecular mechanisms underlying RIF, while these findings of the complexity of miRNA-mediated regulatory networks suggest a need for further in-depth investigations. 

## 5. Conclusions

MicroRNAs (miRNAs) play a crucial regulatory role at different stages of implantation, influencing endometrial receptivity, embryonic development, and the communication between the embryo and endometrium. We anticipate that the two recently identified genes, *AHCYL2* and *BVES*, along with their presumed regulator, miR-665, will serve as markers for identifying RIF disease in Korean women.

## Figures and Tables

**Figure 1 genes-15-00244-f001:**
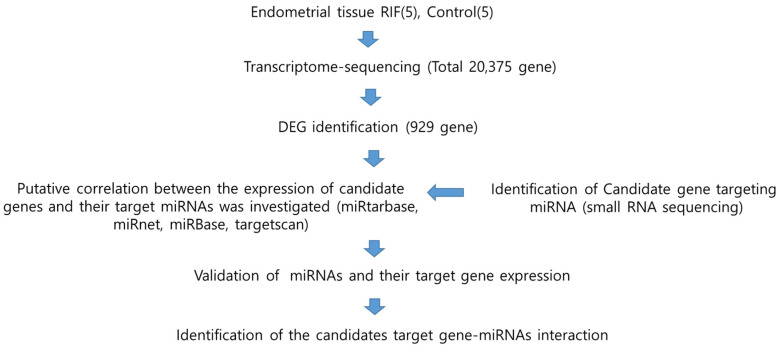
Schematic diagram of the study workflow to determine the association between microRNA and RIF. RIF, recurrent implantation failure; DEG, differentially expressed genes; miRNA, microRNA.

**Figure 2 genes-15-00244-f002:**
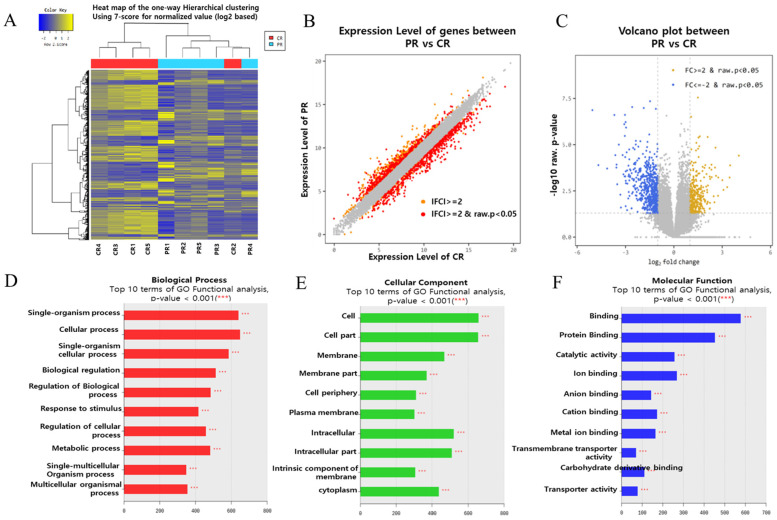
Gene expression profiles of the endometrial tissue of patients with RIF are distinctly different from those of healthy controls. (**A**) RNA sequencing heat map for the 929 most increased or decreased genes in RIF. The color spectrum from blue to yellow indicates low to high expression. (**B**) Scatter plot showing the upregulated and downregulated genes (the red dots, respectively) in the endometrial tissue in the RIF(PR) with respect to the control (CR). Gray dots represent genes with no significant difference. (**C**) Volcano plot (*p*-value vs. linear fold change) of all genes present in RNA sequencing compared between RIF patients and controls. Fold change (FC ≥ 2) with yellow indicates upregulation in RIF; linear fold change (l FC ≥ 2) with blue indicates downregulation in RIF. The X-axis shows the linear FC level, and the Y-axis shows the significance level. RNA sequencing provided a list of 929 differentially expressed genes that overlapped between these two criteria (|Fold-Change| ≥ 2 fold, *p* ≤ 0.05). (**D**–**F**) DAVID functional enrichment analysis, functional Gene Ontology analysis of biological process; (**D**) cellular component; (**E**) molecular function; (**F**) analysis of RIF vs. Control.

**Figure 3 genes-15-00244-f003:**
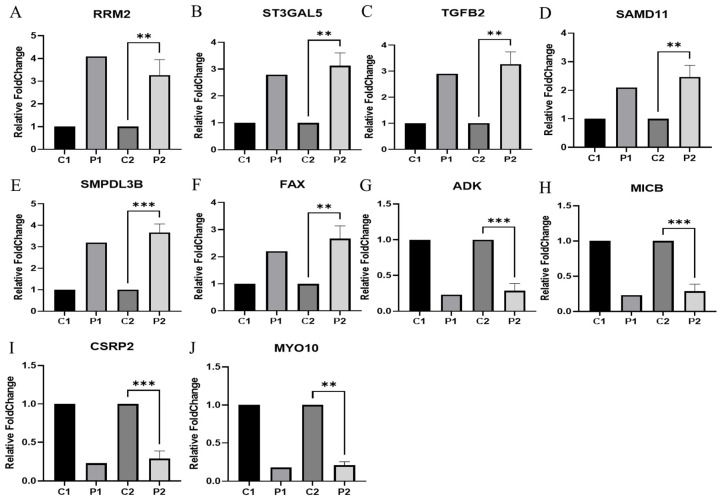
Validation of systemic alteration of genes in these gene sets with real-time RT-PCR. Control and RIF represent the endometrial tissue of healthy control and patients with RIF, respectively. *GAPDH* was used as an internal control, with the relative level of (**A**) *RRM2*, (**B**) *ST3GAL5*, (**C**) *TGFB2*, (**D**) *SAMD11*, (**E**) *SMPDL3B*, (**F**) *FAX*, (**G**) *ADK*, (**H**) *MICB*, (**I**) *CSRP2*, and (**J**) *MYO10* normalized to Gapdh. Abbreviation: C1(NGS data in healthy control), P1(NGS data in RIF). C2 (qRT-PCR in healthy control), P2 (qRT-PCR in RIF). Data represent over three independent experiments with triplicate measurements. ** *p* < 0.05, *** *p* < 0.01.

**Figure 4 genes-15-00244-f004:**
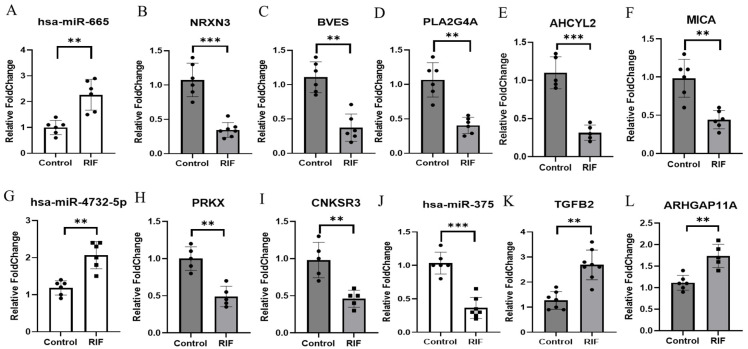
Inverse correlation of miRNAs and their target genes in RIF. (**A**) Expression of has-miR-665 was detected using real-time RT-PCR in RIF and control. U6 snRNA was used as an internal control, with the relative level of has-miR-665 normalized to U6. (**B**–**F**) Expressions *of NRXN3*, *BVES*, *PLA2G4A*, *ACHYL2*, and *MICA* were detected using real-time RT-PCR in RIF and control. (**G**) Expression of has-miR-4732-5p was detected using real-time RT-PCR in RIF and control. U6 snRNA was used as an internal control, with the relative level of has-miR-4732-5p normalized to U6. (**H**,**I**) Expressions of *PRKX* and *CNKSR3* were detected using real-time RT-PCR in RIF and control. (**J**) Expression of has-miR-375 was detected using real-time RT-PCR in RIF and control. U6 snRNA was used as an internal control, with the relative level of has-miR-375 normalized to U6. (**K**,**L**) Expressions of *TGFB2* and *ARHGAP11A* were detected using real-time RT-PCR in RIF and control. All data represent over three independent experiments with triplicate measurements. ** *p* < 0.05, *** *p* < 0.01.

**Figure 5 genes-15-00244-f005:**
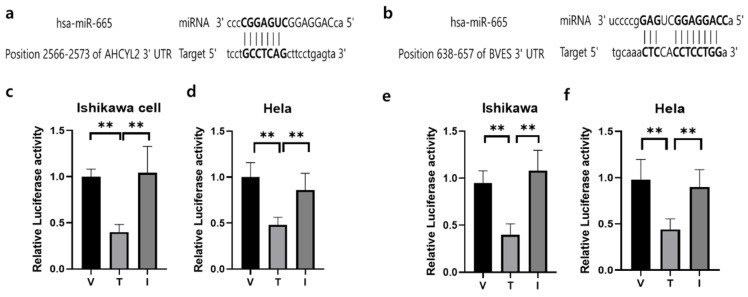
The outcomes from a dual luciferase reporter assay were obtained for miRNAs and their corresponding target genes. (**a**) Prediction of 3′-UTR sequences of *AHCYL2* genes with their binding sites for miR-665, respectively. (**b**) Prediction of 3′-UTR sequences of *BVES* genes with their binding sites for miR-665, respectively. (**c**) Dual luciferase assay results of miR-665 and its target 3′-UTR of *AHCYL2* in Ishikawa cells (**d**) and in Hela cells. (**e**)Dual luciferase assay results of miR-665 and its target 3′-UTR of BVES in Ishikawa cells (**f**) and in Hela cells. Independent miRNA was used as a V:pmiRGLO-3′UTR of *AHCYL2* or pmiRGLO-3′UTR of *BVES*, T: treatment of miR-665 and I: Inhibitor of miR-665. Significant differences between the miRNA target group and inhibitor treatment group are indicated by ** *p* < 0.05.

**Table 1 genes-15-00244-t001:** Sequence of primers used for RT-PCR studies.

Gene	Primer Forward	Primer Reverse
RRM2	aggacattcagcactgggaa	ccatagaaacagcgggcttc
ST3GAL5	tgagaaggcccagcttgtta	ggcaaacttgggacgacatt
TGFB2	acaaaatagacatgccgccc	agggtctgtagaaagtgggc
SAMD11	cgaatatcctccccggtgaa	tcatgatacggatgtgcgga
SMPDL3	ctgcaccttgaccctgacta	ctgcctctcccagtttctca
FAX	ttagtggtggtggtcccttc	gtggcgtcaagagtggaaag
ADK	aatgagcagccaacaggaac	tttcagaagcatggtgagcc
MICB	acacagagaccgaggacttg	gtctgagctctggaggactg
CSRP2	accagagagtgttcagcctc	ccacactttgcacatcggaa
MYO10	agtgcatcctcatcagtggt	catgatggggctgctttcaa
Gapdh	AGGTCGGAGTCAACGGATTT	ATCTCGCTCCTGGAAGATGG

**Table 2 genes-15-00244-t002:** Sequence of primers used for RT-PCR studies of miRNAs and their target genes in RI.F.

Gene/miRNA	Primer Forward	Primer Reverse
NRXN3	ctggttcctgttctggggat	ccaaagatgtacgtagcgcc
BVES	cactctctaccgatgtgcct	gcagcataagtttggccctt
PLA2G4A	tggctcccgacttatttgga	tcatcatcactgtccgagct
AHCYL2	gtcgctctttgtctcgttcc	tcttcgtccaaactctgcct
MICA	caggtcctggatcaacaccc	tggagccagtggacccaa
PRKX	taagctcacggactttgggt	ggaaaccccgaaagcatctc
CNKSR3	actttactcctgctcccctg	ggagagtagggaacagcagg
TGFB2	acaaaatagacatgccgccc	agggtctgtagaaagtgggc
Gapdh	AGGTCGGAGTCAACGGATTT	ATCTCGCTCCTGGAAGATGG
ARHGAP11A	gaaagtgttggttggcgact	ccaactagtcgctctggagt
has-miR-665	CAGGAGGCTGAGGCCCCT	universal primer *
has-miR-4732-5p	TAGAGCAGGGAGCAGGAAGCT	universal primer *
has-miR-375	GTTCGTTCGGCTCGCGTGA	universal primer *
RNU6B (RNU6-2)	ACGCAA ATT CGT GAA GCG TT	universal primer *

Universal primer *: miRCURY LNA SYBR Green PCR Kits (Qiagen).

## Data Availability

The data presented in this study are available on request from the corresponding author.

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
