# Peer review of "miR-665-Mediated Regulation of AHCYL2 and BVES Genes in Recurrent Implantation Failure"

_genes, 2024, doi:10.3390/genes15020244_

Round 1

Reviewer 1 Report

Comments and Suggestions for Authors

The authors of the study "mir665-mediated AHCYL2 and BVES Gene Regulation in Endometrial Dysfunction: Implications for Recurrent Implantation Failure" SH Cho et al dealt with an important issue of female fertility. Their study is very well designed and executed with sound results and conclusions. Nevertheless the title referring to Gene regulation in endometrial dysfunction is somewhat misleading as endometrial dysfunction is mostly used for dysfunctional uterine bleeding. My opinion is that the title should reflect the core of the study, which is the Recurrent Implantation Failure and change to the simpler "mir665-mediated AHCYL2 and BVES gene regulation in recurrent implantation failure".  

Comments on the Quality of English Language

The quality of the English Language is good and the text is sound and comprehensive.  

Author Response

â– The authors of the study "mir665-mediated AHCYL2 and BVES Gene Regulation in Endometrial Dysfunction: Implications for Recurrent Implantation Failure" SH Cho et al dealt with an important issue of female fertility. Their study is very well designed and executed with sound results and conclusions. Nevertheless the title referring to Gene regulation in endometrial dysfunction is somewhat misleading as endometrial dysfunction is mostly used for dysfunctional uterine bleeding. My opinion is that the title should reflect the core of the study, which is the Recurrent Implantation Failure and change to the simpler "mir665-mediated AHCYL2 and BVES gene regulation in recurrent implantation failure".  

⇒ Thank you very much for your helpful comments. According to your suggestion, we have revised the Title as follows: “miR-665-mediated Regulation of AHCYL2 and BVES Genes in Recurrent Implantation Failure”

Reviewer 2 Report

Comments and Suggestions for Authors

2.2 Subjects - please provide more information about the participants included (what criteria for high quality embryos, how many transfers, infertility diagnosis, endometrial preparation; in the control group were people with infertility excluded)

Endometrial tissue was collected in natural cycles. Were all embryo transfers also in natural cycles? What is the impact of exogenous hormones on the endometrium that is common in embryo transfers?

Line 113 - remove either unless there is another condition

Clarify need for figure 2b and 2c. It seems to be same information in different presentation.

Define ET

Figure 3 - how many patient samples are included in the data. Are the same samples used for NGS and qRT-PCR?

Define DEmiRs

Line 318: Small RNA-sequencing... This is not a complete sentence.

Section 3.4 - Most of this is the legend for supplementary figure and does not provide additional information.  Needs to be revised

Description of Table 1 (line 339) does not match what is included in the table

Figure 5 - mutant vectors of the putative binding site were not shown, but described in the legend. This would be useful to show.

The genes further analyzed were selected because of importance in angiogenesis and cell adhesion, but these processes/functions were not identified the the GO analysis. Please explain.

While AHCYL2 and BVES are altered in RIF patients, what can be done to change expression? Would this impact implantation? Are there compensatory mechanisms?

Comments on the Quality of English Language

Overall, English is fine. There are some minor edits that can be made.

Author Response

â– 2.2 Subjects - please provide more information about the participants included (what criteria for high quality embryos, how many transfers, infertility diagnosis, endometrial preparation; in the control group were people with infertility excluded)

⇒ Thank you very much for your helpful comments and suggestions. We have revised and added the sentence as follows in Section 2.2 subjects:

The patients were enrolled from June 2017 through May 2018 at the Department of Obstetrics and Gynecology of CHA Bundang Medical Center (Seongnam, South Korea).

RIF was defined as failure to achieve pregnancy after two or more completed fresh IVF-embryo transfer (IVF-ET) cycles with more than 10 cleaved embryos. Serum human chorionic gonadotrophin concentrations were less than 5 mIU/mL 14 days after embryo transfer. All transferred embryos were examined by the embryologist before transfer and judged to be of good quality. The criteria for identifying high-quality embryos in this study were determined by assessing specific parameters, such as morphological characteristics, developmental stage and genetic normalcy. Hormonal causes of infertility, including hyperprolactinemia, luteal insufficiency, and thyroid disease, were excluded by measuring the concentrations of prolactin, thyroid stimulating hormone, free T4, follicle-stimulating hormone, luteinizing hormone, and progesterone in peripheral blood. Lupus anticoagulant and anticardiolipin antibodies were examined to identify the autoimmune diseases lupus and antiphospholipid syndrome, respectively. A thrombotic cause of RIF, namely thrombophilia, was evaluated by protein C and protein S deficiencies and by the presence of anti-a2 glycoprotein antibodies. Semen analysis, karyotyping, and oestradiol, testosterone, follicle-stimulating hormone, and luteinizing hormone assays, were performed for male partners.

â– Endometrial tissue was collected in natural cycles. Were all embryo transfers also in natural cycles? What is the impact of exogenous hormones on the endometrium that is common in embryo transfers?

⇒ Were all embryo transfers also in natural cycles?

“While the endometrial tissue in our study was harvested according to the natural cycle, embryo transfers can be performed in various ways. In some cases, natural cycles are used, but assisted reproductive technologies often involve controlled ovarian stimulation to optimize the chances of successful implantation. The choice of cycle type depends on individual patient characteristics, treatment protocols, and clinical considerations.”

⇒ What is the impact of exogenous hormones on the endometrium that is common in embryo transfers?

“The use of exogenous hormones in embryo transfers has a significant impact on the endometrium. Hormonal medications such as gonadotropins, estrogen, and progesterone are commonly employed to optimize the uterine environment for embryo implantation. These hormones help control the development of follicles, regulate the timing of ovulation, and support the thickening of the endometrial lining. Specifically, estrogen is administered to stimulate endometrial growth, while progesterone is introduced to mimic the hormonal conditions following ovulation, promoting receptivity for embryo implantation. These exogenous hormones aim to create an ideal environment within the uterus, increasing the chances of successful implantation and subsequent pregnancy during assisted reproductive procedures.”

â– Line 113 - remove either unless there is another condition

⇒ Thank you for your kind comments. We have revised a sentence as follows:

“These biopsied samples were promptly transported to the research laboratory and snap-frozen at -80°C for RNA extraction.”

Please, see the revised sentence!

â– Clarify need for figure 2b and 2c. It seems to be same information in different presentation.

⇒ We apologize for any confusion caused by our explanation of Figure 2. We have added labeling (A~F) to Figure 2. Please, see the revised Figure 2!

â– Define ET

⇒ Thank you for your kind comments. We have added a sentence as follows:

â– 3.3. Validation of Differential Expressed Genes

Real Time PCR (RT-qPCR) was performed using Endometrial tissue (ET) to verify the gene RNA-sequencing results for selected DEGs. Of the Endometrial tissue - Differential Expressed Genes (ET-DEGs), the expression levels of random selected 10 genes were statistically significantly different [adjusted p ≤0.05] between RIF and the corresponding controls (Figure 3).

â– Figure 3 - how many patient samples are included in the data. Are the same samples used for NGS and qRT-PCR?

⇒ Thank you for your kind comments. In this study, qRT-PCR was conducted utilizing identical samples employed for Next-Generation Sequencing (NGS) data acquisition.

â– Define DEmiRs

⇒ Thank you for your kind comments. We have added a full name as follows:

â– 3.4. Identification of Differentially Expressed miRNAs

Differentially Expressed miRNAs (DEmiRs) ~

â– Line 318: Small RNA-sequencing... This is not a complete sentence.

⇒ Thank you for your kind comments. We have added a sentence as follows:

“A small RNA-sequencing heat map was constructed to depict the expression profiles of the top 24 genes exhibiting the most significant increases or decreases in response to RIF”

Please see 3.4. Identification of Differentially Expressed miRNAs

â– Section 3.4 - Most of this is the legend for supplementary figure and does not provide additional information.  Needs to be revised

⇒ Thank you for kind comments. According to your suggestion, we added and revised the following sentence to Section 3.4:

We performed miRNA sequencing in the endometrial tissue samples derived from patients with RIF (n=5) and successfully clinical pregnancy (n=5) to identify the differential miRNAs. To uncover previously unidentified miRNAs within our sequencing dataset, we aligned the clean miRNA reads with the human genome and employed miRDeep2 (version 2.0.08), an algorithm grounded in the miRNA biogenesis model. A total of 1378 miRNAs were analyzed through small RNA sequencing, and among them, 98 up-regulated miRNAs and 88 down-regulated miRNAs were selected based on significance tests (significant RandFold p-values, p < 0.05). Twenty-four miRNAs (14 up-regulated, 10 down-regulated) met the criteria of fold change (|fc| ≥ 2) and raw p < 0.05 using DESeq2. We used the online tools miRTarBase and miRnet to pinpoint experimentally validated targets supported by strong evidence. From these findings, we identified seventeen miRNAs demonstrating a negative correlation with the candidate genes (Supplementary Figure S1).

We revised the following sentence to Section 3.4:

As a result, we identified a set of differentially expressed miRNAs (DEmiRs) comprising 10 up-regulated (hsa-miR-3667-3p, hsa-miR-6858-3p, hsa-miR-216a-3p, hsa-miR-410-5p, hsa-miR-106a-3p, hsa-miR-6814-3p, hsa-miR-1224-3p, hsa-miR-4732-5p, hsa-miR-129-2-3p and hsa-miR-665) and 7 down-regulated (hsa-miR-30d-5p, hsa-miR-3188, hsa-miR-375, hsa-miR-6894-5p, hsa-miR-1913, hsa-miR-4743-5p and hsa-miR-3663-3p) with demonstrating high reliable miRNAs in RIF (Supplementary Table 3).

Please see 3.4. Identification of Differentially Expressed miRNAs!

â– Description of Table 1 (line 339) does not match what is included in the table

⇒ We apologize for any confusion caused by our explanation of Table I.

We added a table by changing Table I in the original manuscript (line 339) to Supplementary Table 4. Please see the revised Supplementary table 4! (3.5. Inverse Correlation of miRNAs and Putative Target Genes in the ET of Control and RIF)

â– Figure 5 - mutant vectors of the putative binding site were not shown, but described in the legend. This would be useful to show.

⇒ We apologize for any confusion caused by our explanation of Figure 5.

We have revised a sentence as follows: “Significant differences between the miRNA target group and inhibitor treatment group are indicated by ** p < 0.05.”

â– The genes further analyzed were selected because of importance in angiogenesis and cell adhesion, but these processes/functions were not identified the the GO analysis. Please explain.

⇒ Thank you very much for your helpful comments

Gene Ontology (GO) analysis serves as a valuable biological tool that characterizes the features of genes derived from Next-Generation Sequencing (NGS) data on a genome-wide scale. This makes it an essential analytical method for researchers aiming to unravel the functional tendencies of genes during the early phases of NGS data analysis. However, this method has limitations in reflecting the detailed characteristics of specific diseases. Therefore, there is a need to track the characteristics of genes associated with diseases such as RIF for research purposes. We focused on angiogenesis and cell adhesion-related genes, which are known to be significantly associated with RIF. Based on the obtained NGS data, we separately tracked genes related to angiogenesis and cell adhesion.

To this end, we first screened 24 significant miRNAs and 71 candidate genes targeting them from small RNA sequencing data using miRTarbase, mirnet, miRbase, and targetscan (Supplementary Table 4). Among them, we selected NRXN3, TGFBR2, KSR2, AHCYL2, PLA2G4A, MICA, PRKX, CNKXR3, BVES, TGFB2, and ARHGAP11A, which are associated with angiogenesis and cell adhesion. Finally, has-mIR-665 (targeting AHCYL2, PLA2G4A, MICA, BVES, NRXN3), has-mIR-4732-5p (targeting PRKX, CNKXR3), and has-miR-375 (targeting TGFB2, ARHGAP11A) were selected to target these candidate genes.

â– While AHCYL2 and BVES are altered in RIF patients, what can be done to change expression? Would this impact implantation? Are there compensatory mechanisms?

⇒ Thank you very much for your helpful comments

To reverse the decreased expression of genes such as AHCYL2 and BVES, it is possible to increase the expression of AHCYL2 or BVES or to use an inhibitor of miR-665, which regulates these genes.

In this case, direct injection of miR-665 inhibitors or overexpression vectors of AHCYL2 or BVES into the endometrium could be considered as a potential approach for gene therapy of RIF, but it is important to note that the feasibility of this approach depends on many factors. Delivery systems, such as viral vectors or nanoparticles, must be designed to avoid potential side effects and ensure delivery to the intended cells.

In addition, the safety profile of the chosen delivery method and the potential adverse effects of inhibiting miR-665 or overexpressing AHCYL2/BVES should be thoroughly evaluated. A comprehensive understanding of the molecular mechanisms involved in the regulation of implantation-related genes and the effects of miR-665 is essential for this, and we believe that further studies should be conducted.

Altered expression of miR-665 and its downstream effects on the BVES and AHCYL2 genes may actually affect the implantation process. Both the BVES and AHCYL2 genes are associated with angiogenesis, an important process for successful pregnancy. Therefore, we expect that restoring the expression of these genes will affect the development and function of endometrial blood vessels, potentially positively affecting embryo implantation.

As for compensatory mechanisms, biological systems often have mechanisms to maintain balance and homeostasis. In response to changes in gene expression, cells may activate compensatory pathways to mitigate the impact of these alterations. For example, other miRNAs or regulatory factors might come into play to counterbalance the effects of increased miR-665 expression.

We expect that further studies on angiogenesis-associated genes such as AHCYL2 and BEVES in RIFs will enable us to identify ways to regulate these compensation pathways or directly target dysregulatory components to restore normal function and support successful transplantation

Reviewer 3 Report

Comments and Suggestions for Authors

Comments about the manuscript:

“miR-665-Mediated AHCYL2 and BVES Gene Regulation in Endometrial Dysfunction: Implications for Recurrent Implantation Failure”

Despite advances in assisted reproduction techniques, many infertile couples continue to experience repeated therapeutic failures. The objective of this study was to highlight which mRNA targets were affected by deregulated miRNAs in recurrent implantation failures. To do this, the authors analyzed mRNA and miRNA expression profiles in women with no problem reproductive function and women with recurrent implantation failure, by performing transcriptome and small RNA sequencing on endometrial tissues. The involvement of deregulated genes has thus been confirmed, notably significant changes in the expression of genes linked to angiogenesis and cell adhesion.

This important work that brings meaningful and useful results. However, the manuscript requires some improvements.

Page 1, line 33. Write “BVS” in full and as an abbreviation (this is the first time the abbreviation appears in the abstract).

General: Given the large number of abbreviations, it would be useful to add a list of all abbreviations (in alphabetical order) at the beginning or end of the text.

Page 3, lines 112-113. “These biopsied samples were promptly transported to the research laboratory, where they were either snap-frozen at -80°C for RNA extraction”: "where they were either frozen at -80°C for RNA extraction" or... what? complete the sentence or delete “either or”.

Page 4, line 143, 144-145. “according to the manufacturer’s instructions” is not sufficient for a scientific paper. Briefly describe the technique.

Page 5, lines 176, 192. The same : “according to the manufacturer’s protocol” is not sufficient for a scientific paper. Briefly describe the technique.

Page 6, line 258 and others: use italics to write the names of the genes.

Page 9, line 340. “ As a result, we identified 17 specific miRNAs targeting 71 candidate genes (Table I).”: I didn't understand the relationship between Table 1 and this paragraph. Please explain on the table.

Page 10, figure 4Check the chronology of the figures (a, b, c...): there seems to be a confusion between the letters.

Page 11, figure 3. There seems to be a problem with the presentation of this figure: what are parts a, b, c, d, e, f? Please check and correct.

Page 14, line 516. Use italics to write Homo sapiens.

Supplementary figures: Indicate the subdivisions a, b, c... on the figure. Explain what the third page of this supplementary file is.

Supplementary Tables 1-4 are not referred to in the text.

Author Response

â– Page 1, line 33. Write “BVS” in full and as an abbreviation (this is the first time the abbreviation appears in the abstract). ⇒ Thank you for your kind comments.

Based on my understanding, the initial mention of Blood vessel epicardial substance (BVES) in the abstract can be found in line 31. I would appreciate it if you could check line 31!

â– General: Given the large number of abbreviations, it would be useful to add a list of all abbreviations (in alphabetical order) at the beginning or end of the text.

⇒ Thank you very much for your helpful comments and suggestions.

"Anticipating potential difficulties in integrating abbreviations into the main text, I took some time to deliberate and ultimately opted to include a list of abbreviations in the supplementary file. I hope for your understanding on this matter."

I would be grateful if you could kindly review the supplementary file!

â– Page 3, lines 112-113. “These biopsied samples were promptly transported to the research laboratory, where they were either snap-frozen at -80°C for RNA extraction”: "where they were either frozen at -80°C for RNA extraction" or... what? complete the sentence or delete “either or”.

⇒ We apologize for any confusion caused by our explanation of the sentence.

We have revised a sentence as follows: “These biopsied samples were promptly transported to the research laboratory and snap-frozen at -80°C for RNA extraction.”

â– Page 4, line 143, 144-145. “according to the manufacturer’s instructions” is not sufficient for a scientific paper. Briefly describe the technique.

⇒ Thank you for your kind comments. We have added a sentence as follows:

“1 μg total RNA was ligated with 3′ and 5′ adaptors followed by reverse transcription with Reverse Transcriptase and amplification of the small RNA sequences during subsequent steps. This step converts RNA into complementary DNA (cDNA), preserving sequence information. The cDNA libraries are then amplified in a 15-cycle. The small RNA fraction, comprising the 130–150 base pair region, was excised from a 6% polyacrylamide gel post-electrophoresis, subjected to ethanol precipitation, and quantified.”  Please see 2.4. Small-RNA Sequencing!

â– Page 5, lines 176, 192. The same : “according to the manufacturer’s protocol” is not sufficient for a scientific paper. Briefly describe the technique.

⇒ Thank you for your kind comments. We have added a sentence as follows:

“Thaw all RNA on ice. Thaw 10x miScript Nucleics Mix and either 5x miScript Hiflex (or Hispec) Buffer at room temperature (15–25ºC). Mix each solution by mixing well the tubes. And then Centrifuge briefly to collect residual liquid from the sides of the tubes and then store on ice. Add RNA to each tube containing reverse-transcription master mix (total 20 microliter). Incubate at 37°C for 60 minutes for amplification.”

Please see 2.6. and 2.7!

â– Page 6, line 258 and others: use italics to write the names of the genes.

⇒ Thank you for your kind comments. We've revised the gene names to be italicized including page 6, line 258 as follows:

“This list included top10 genes (GDF10, S100A9, ASTN1, COL26A1, CYP24A1, SCAR-NA13, PDZK1P1, PDZK1, FAM129C and USP32P1) ~…………… and Top10 genes (SLC24A4, PHF24, XDH, HLA-DOB, MEGF10, TRPM6, KCNG1, PKHD1L1, LOC101928150 and CAPN6) ~” Please see revised manuscript!

â– Page 9, line 340. “ As a result, we identified 17 specific miRNAs targeting 71 candidate genes (Table I).”: I didn't understand the relationship between Table 1 and this paragraph. Please explain on the table.

⇒ We apologize for any confusion caused by our explanation of Table I.

We added a table by changing Table I in the original manuscript (line 339) to Supplementary Table 4. Please see the revised manuscript and supplementary table 4!

â– Page 10, figure 4Check the chronology of the figures (a, b, c...): there seems to be a confusion between the letters.

⇒ Thank you for your helpful comments and suggestions. We revised Figure 4 and the letters. Please see the revised Figure 4!

â– Page 11, figure 3. There seems to be a problem with the presentation of this figure: what are parts a, b, c, d, e, f? Please check and correct.

⇒ Thank you for your helpful comments and suggestions. We revised Figure 3 and the letters. Please see the revised Figure 3!

â– Page 14, line 516. Use italics to write Homo sapiens.

⇒ Thank you for your kind comments. We have revised a sentence as follows: “hsa-miR-4732-5p is an identified microRNA (miRNA) found in Homo sapiens,~

â– Supplementary figures: Indicate the subdivisions a, b, c... on the figure. Explain what the third page of this supplementary file is.

⇒ We apologize for any confusion caused by the previous information. We have deleted the third page of the supplementary figure.

â– Supplementary Tables 1-4 are not referred to in the text.

⇒ Thank you for your kind comments. We have included a supplementary table within each section of the results as follows:

3.1. Gene Expression Profiles of the Endometrial Tissue with RIF Are Distinctly Different from Those of Healthy Control: (Supplementary Tables 1 and 2)

3.4. Identification of Differentially Expressed miRNAs: (Supplementary Tables 3)

3.5. Inverse Correlation of miRNAs and Putative Target Genes in the ET of Control and RIF: (Supplementary Tables 4)

3.6. AHCYL2 and BVES Are Direct Target of miR-665 in Ishikawa Cells: (Supplementary Tables 5)

"I would appreciate it if you could review the Results section."
